# Nurses’ Awareness of and Current Approaches to Oral Care in a Community Hospital in Japan: A Longitudinal Study of Dental Specialists’ Interventions

**DOI:** 10.3390/healthcare11172425

**Published:** 2023-08-30

**Authors:** Takashi Koike, Ryuichi Ohta, Yuhei Matsuda, Chiaki Sano, Takahiro Kanno

**Affiliations:** 1Department of Oral and Maxillofacial Surgery, Unnan City Hospital, Unnan 699-1221, Japan; 2Department of Oral and Maxillofacial Surgery, Faculty of Medicine, Shimane University, Izumo 693-8501, Japan; yuhei@med.shimane-u.ac.jp (Y.M.); tkanno@med.shimane-u.ac.jp (T.K.); 3Department of Community Care, Unnan City Hospital, Unnan 699-1221, Japan; ryuichiohta0120@gmail.com; 4Department of Community Medicine Management, Faculty of Medicine, Shimane University, Izumo 693-8501, Japan; sanochi@med.shimane-u.ac.jp

**Keywords:** mouth, care, nurses, dentistry, community hospital, burden, intervention

## Abstract

**Objective:** This study aimed to increase nurses’ interest and priority in oral care by implementing educational interventions, such as teaching oral care methods suitable for each patient in clinical settings, over a period of one year. **Materials and Methods:** This study included all 150 nurses working in Unnan City Hospital in Japan who answered a questionnaire comprising 19 questions regarding awareness, actual implementation status of oral care provided, burden, and involvement with oral and maxillofacial surgery department of the hospital, along with participants’ characteristics. The rate of interest in learning, need for oral care, time spent in oral care, and oral health-related caregiver burden index (OHBI) score were compared between pre- and post-intervention groups. **Results:** The number and rate of valid questionnaires were 136 and 90.7%, respectively. The mean years of clinical experience were 19.3 ± 12.5 years; 93.4% of the nurses were women. After the interventions by dental specialists, the nurses’ level of interest in and priority to oral care were significantly higher than those before the interventions (*p* < 0.001), regardless of nurses’ background, such as age, gender, or years of experience. However, the “burden” did not statistically decrease. **Conclusions:** This study shows that dental specialists succeeded in significantly increasing nurses’ interest in and priority to oral care by intervening in clinical practice but failed in decreasing nurses’ burden of oral care. In the future, we would like to investigate the problems that hinder the reduction of the sense of burden, reduce the burden of nurses’ oral care, and improve oral care.

## 1. Introduction

The oral cavity is a gateway into the respiratory and digestive organs and has important functions, such as chewing, swallowing, and articulation. However, leaving the oral environment in poor conditions affects not only the oral cavity but also other organs, including respiratory organs, which aggravates patients’ general condition. In addition, inadequate oral care may worsen the oral environment and general condition, and may damage the oral cavity [1,2,3,4]. Oral care controls bacterial proliferation and ensures oral hygiene to prevent airway infections. Oral care also maintains and improves oral functions such as eating and swallowing. Therefore, the importance of oral care has been re-evaluated in recent years [1,2]; oral care improves patients’ quality of life (QOL) and activity of daily living (ADL), and contributes significantly to patients’ independence. As a result, the need for oral care is now more strongly recognized in medical, health, and welfare settings [3,4]. In preventing aspiration pneumonia in elderly patients, oral hygiene management has also become important for whole-body management [1,2,3,4]. Oral care is one of the basic nursing skills. In particular, nurses have provided oral care to patients who are unable to perform it themselves. However, while awareness of oral care is increasing, nurses are too busy to provide adequate oral care to patients in many hospitals and facilities [5,6,7].

In the previous studies, ward nurses were surveyed regarding their interest in oral care, actual oral care, such as usual oral care time and frequency, and their self-assessment, such as satisfaction with the usual oral care provided to patients [5,6,7,8]. Hospitalized patients for whom ward nurses provide oral care are at high risk of infection, since they have a decreased resistance to infection due to their conditions and treatments. Patients with multiple health problems, such as oral dysfunction leading to deterioration of eating and swallowing, have a high risk of aspiration owing to oral care, making it difficult to provide individualized oral care for such patients.

Both routine oral care by ward nurses and professional oral care by dental specialists such as dentists and dental hygienists are essential for providing adequate oral care to patients. It is important for nurses to understand the current status, knowledge, and awareness of oral care and to identify the problems they face in clinical practice. In our previous study, we conducted a questionnaire survey on nurses to promote an environment for better oral care for hospitalized patients [8]. We performed a statistical analysis of factors associated with the burden of oral care. Our results showed that there was no association between “being highly interested in oral care” and “giving high priority to oral care” and the burden of oral care. This suggested that nurses who were highly interested in oral care or gave high priority to oral care were less likely to perceive the burden of oral care [8]. Therefore, in this study, we performed educational interventions, including direct instructions in clinical practice over a period of one year, with the goal of increasing nurses’ interest in and priority to oral care and improving the oral care provided to hospitalized patients. Thereafter, we conducted a questionnaire survey to assess the results of changes in nurses’ awareness of oral care during this one-year period.

## 2. Materials and Methods

### 2.1. Participants

We surveyed all 150 nurses working in Unnan City Hospital, which is a general hospital with 281 beds and 15 departments at present, and is also a community core hospital in Unnan City. Before the opening of the oral and maxillofacial surgery department at the hospital in September 2019, nurses provided all oral care to hospitalized patients. Unnan City is located in the mountainous area of Shimane Prefecture in western Japan, with a population of approximately 36,000 and an aging population rate of 40.2%. This area is one of the most rapidly aging and shrinking populations in Japan.

### 2.2. Study Design

This was a longitudinal study.

### 2.3. Survey Methods

Following our previous study, we provided nurses with educational interventions, such as direct on-site instruction, over a period of approximately one year. For the purpose of investigating changes after the interventions, we conducted an anonymous questionnaire survey on oral care to investigate changes after the intervention. The intervention was performed by a dentist with specialized qualifications in oral and maxillofacial surgery and two dental hygienists with more than 20 years of clinical experience. We intervened when requested by the attending physicians and nurses for inpatients, determined the frequency and time of intervention, and explained the oral conditions and appropriate oral care methods to the nurses for each patient. To ensure the reliability and validity of the questions, we developed a questionnaire based on previous studies [5,6,7,8,9,10,11,12,13,14,15]. The questionnaire consisted of 19 questions similar to our previous research and included questions on awareness of oral care, actual oral care performed by nurses themselves, the burden of oral care, presence or absence of involvement with the oral and maxillofacial surgery department of the hospital, such as receiving direct guidance and instructions on oral care in clinical settings, and basic attributes such as sex, age, and department [8]. We explained the purpose, significance, and methods of this study to the hospital’s Director of Nursing and obtained permission to conduct the questionnaire survey. The questionnaire was accompanied by a request document and a consent form. The request document explained the purpose and methods of this study to nurses. It mentioned that participation in this study was voluntary, participants had a right to refuse participation, and nonparticipation would not cause any disadvantage to the participants. The data obtained are used for research purposes only, and privacy was respected. We distributed questionnaires to all nurses through the Director of Nursing in early December 2022. Only those nurses who agreed to participate in this study responded to the questionnaire, which was collected three weeks after distribution. The questionnaires were promptly discarded after this study was completed. This study was approved by the Ethics Committee of Unnan City Hospital (approval number: 20200021).

### 2.4. Data Collection Procedures

A total of 150 questionnaires were distributed to the nurses, and 142 responses were collected, with a response rate of 94.7%. Incomplete responses, such as those in which some items were not filled in or were filled in incorrectly, were considered invalid. Finally, 136 valid responses were obtained, yielding a valid response rate of 90.7%. Thereafter, the awareness of oral care, burden of oral care, involvement with the oral and maxillofacial surgery department of the hospital, and basic attributes of 136 nurses’ awareness of oral care, the burden of oral care, involvement with the oral and maxillofacial surgery department of the hospital, and basic attributes were tabulated, and their percentages were calculated. In addition, we examined the number of oral care sessions per day and the duration of oral care per session by each ward nurse. For this, if there was a range of values in the responses, the maximum value was used. We examined the burden of oral care using the Oral Health-related Caregiver Burden Index (OHBI) as a scale of burden [16]. OHBI contains nine question items on the following four domains (technique-related burden, service-related burden, existential burden, and risk-related burden) and one question on the overall burden. Using the responses to each question, the burden the nurses perceived was scored based on the following five options, “never”, “rarely”, “sometimes”, “quite frequent”, and “nearly always”. In addition, in the item about the involvement with the oral and maxillofacial surgery department, additional questions were asked to the nurses who answered that they had ever received guidance and instructions from the oral and maxillofacial surgery department. Those questions were whether the problems were solved and whether there was a change in their awareness of oral care.

### 2.5. Statistical Analysis

Student’s *t*-test was performed on parametric data, and the Mann–Whitney U test was performed on non-parametric data. Numerical variables were dichotomized as follows: learning interest (strong or a little = 1; neutral, no interest, or no idea = 0); need for oral care (strong or a little = 1, neutral, no interest, no idea = 0); and priority of oral care (very high or relatively high = 1, not high, not relatively high, or no idea = 0). In addition, the actual implementation status of oral care and OHBI score was dichotomized by the median. The rate of learning interest, the necessity of oral care, the priority of oral care, time spent on oral care, and OHBI scores were compared between pre- and post-intervention groups. The χ^2^ test was used to test the statistically significant differences between the two groups for high-interest, high-need, and high-priority groups, actual oral care time, and total OHBI scores. Cases with missing data were eliminated from the analysis. Statistical significance was defined as a *p* < 0.05. All statistical analysis was performed using EZR (Saitama Medical Center, Jichi Medical University, Saitama, Japan), a graphical user interface for R (The R Foundation, Vienna, Austria).

## 3. Results

### 3.1. Participant Characteristics

The mean years of clinical experience of the participants was 19.3 ± 12.5 years, and 93.4% of the nurses were women. Of all the nurses, 80.1% were ward nurses (Table 1).

### 3.2. Awareness of Oral Care

Of all the nurses, 91.9% answered “strong” or “a little” to the question, “Are you interested in oral care for hospitalized patients?” and 99.3% answered “strong” or “a little” to the question, “Do you think that oral care is necessary for hospitalized patients?” Similar to the level of interest, a large number of nurses felt that oral care was necessary. In addition, 59.6% of the nurses answered that the priority to oral care is “very high” or “relatively high” compared to care for other areas of the body. About half of the nurses considered that they should give a high priority to oral care. Regardless of nurses’ background, such as age, gender, or years of experience (Table 2), the nurses’ level of interest in and priority to oral care were significantly higher after the interventions by dental specialists as compared to those before the interventions (*p* < 0.001) (Table 3).

### 3.3. Actual Implementation Status of Oral Care

The median duration of oral care sessions for a patient was 5.0 min, and the median number of oral care sessions for a patient was 2.32 per day. The median ideal number of oral care sessions was 3.05 per day. Of all the nurses, 7.6% answered that they were “very much satisfied” or “satisfied” with the oral care they provided.

### 3.4. Burden of Oral Care

The perceived burden of oral care for nurses was scored using OHBI. The burden score of each nurse was calculated by totaling the points. We made a comparison of the median of the score before and after the interventions. There was no statistically significant difference in the median of the burden score before and after the interventions (*p* = 0.593) (Table 3).

### 3.5. Involvement with Oral and Maxillofacial Surgery

Of all the nurses, 36.0% answered that they had received on-site guidance and instructions from the oral and maxillofacial surgery department during the past year. Among them, 83.7% answered that the involvement with the dental staff helped them solve problems, and 87.8% answered that the involvement changed their awareness of oral care. Both of these percentages were quite high. However, the mean score of burden (OHBI) of the nurses who showed involvement with the dental staff (49 nurses) was 24.8, and for those who did not show involvement (87 nurses), it was 25.1, showing no statistical difference between either scores.

## 4. Discussion

In this study, 91.9% of the nurses answered “strong” or “a little” interested in oral care for hospitalized patients. The percentage of nurses with a high level of interest increased after the interventions by the dental specialists compared to those before the interventions. The percentage of nurses who answered that oral care is necessary for hospitalized patients after the interventions was 99.3%, which was very high, and increased compared to 98.5% before the interventions. In addition, 59.6% of the nurses answered that the priority to oral care was “very high” or “relatively high” compared to care for other areas of the body. Compared to 30.6% before the interventions, the percentage of nurses who considered that they should give a high priority to oral care increased significantly. There was no difference in the time taken to perform one oral care session per patient, with a median value of 5.0 before and after the intervention. There was no statistically significant difference in the mean duration before and after the interventions. The mean numbers of oral care sessions for a patient before and after the interventions were 2.4 ± 1.0 and 2.3 ± 0.9 per day, respectively. There was also no statistically significant difference in the number. However, the mean ideal numbers of oral care sessions before and after the interventions were 3.3 ± 0.9 and 3.1 ± 0.7, respectively, which were both higher than the actual number of sessions performed. Of all the nurses, 8.7% and 7.6% were “very much satisfied” or “satisfied” with their oral care before and after the interventions, respectively, which were both low. Similar to our previous study, we assumed that the nurses have a relatively high awareness of oral care, but they may be too busy with their daily tasks to take out much time to provide oral care. Our previous study showed an association between nurses’ “burden” of oral care and their “interest” and “priority”, suggesting that the nurses with higher levels of interest and priority are less likely to perceive the burden. Based on these results, in order to increase nurses’ interest in and priority to oral care, the dental specialists intervened in oral care by providing some direct instructions for the nurses. The instructions included observing the oral cavity together in the clinical practice, pointing out oral care problems, and explaining the need for oral care. As a result of the interventions, we succeeded in achieving a statistically significant increase in interest and priority. However, the “burden” did not statistically decrease. The degree of the present interventions may not have been high enough to reduce the burden; therefore, the present interventions may not have contributed to an increase in the number and duration of oral care sessions or the nurses’ satisfaction. Although there was no significant difference in the median of the burden score before and after the interventions, the score decreased somewhat after the interventions when comparing the score distribution before (15–43) and after (9–39) the interventions. This suggests that raising the degree of interventions may make a significant difference in the future. Of all the nurses, 36.0% had received guidance and instructions from the dental specialists over a period of one year. Among them, 83.7% answered that the interventions solved their problems, and 87.8% answered that their awareness of oral care had changed after the interventions. Although both percentages were high, there was no statistically significant difference in the mean score of burden (OHBI) before and after the interventions. This result suggested that the degree of the present interventions was high enough to improve the nurses’ awareness of oral care, but not to reduce their burden. From these findings, we concluded that we need more practical learning about oral care, such as presenting specific cases and data through lectures and training sessions.

In Japan, a super-aging society, the rate of hospitalized patients aged 65 or older is extremely high [17]. In particular, a community hospital, such as the one in this study, has many super-elderly inpatients [18]. Declines in ADL decreased eating and swallowing functions, and reduced salivary production are often observed in super-elderly persons. Therefore, they are at high risk for deterioration of their oral environment. In addition, many super-elderly inpatients experience a cognitive decline, and oral care of elderly patients with dementia is known to increase the burden on nurses. Actually, many nurses have often complained of struggling to cope with patients with dementia who refuse oral care. In this study, 87.7% of the nurses answered “sometimes”, “quite frequent”, or “nearly always” to the question regarding burden, for example”, I have a hard time because patients resent receiving oral care”. This may be one of the reasons why nurses’ burden did not significantly decrease even after one year of the interventions by the dental specialists. Elderly persons with dementia tend to have oral problems because of decreased motivation for self-care, refusal of assistance, changes in eating behavior, and deterioration of the oral environment due to drug side effects. In addition, due to reduced perception, such patients are less likely to be aware of pain and discomfort symptoms in addition to age-related deterioration of hand movements and eating and swallowing functions. Thus, they face many problems maintaining oral hygiene. It is important to support independence in self-care for patients with dementia as much as possible by respecting their own motivation and abilities. When the quality of their self-care declines with the progression of dementia, assistance in care by caregivers is essential [19]. However, oral care is often difficult in actual clinical practice due to refusal to care or difficulty opening the mouth. Previous studies reported that the refusal behavior of patients with dementia to oral hygiene management appears as a defensive response to stimuli that are negative for the patients, because the oral cavity is an extremely sensitive area. They also mentioned that successful oral care requires reassuring responses to patients [20], such as plain language, eye contact, slow movements, and careful physical contact. In addition, refusal behavior to oral care can be seen even in patients with relatively mild dementia, which can lead to a burden on caregivers [21]. Oral care is an activity of daily living; therefore, not only healthcare professionals but also patients themselves and their caregivers, including family members, need to understand the necessity of oral care so it can be continued at home post-hospital discharge. In the future, we plan to increase opportunities for interventions, such as lectures and practical training sessions by dental specialists, as it will be important for nurses to learn ways to treat elderly patients with dementia through verbal and non-verbal communication, given the high incidence of patients with dementia in community hospitals.

Contrarily, it is argued that oral care education programs in lectures and practical training sessions should not just concentrate on oral care methodology but also on improving oral care quality through problem-solving behavior for individualized oral care [22]. This problem-solving behavior was defined as ward nurses’ behavior to solve patients’ health and life problems, which apparently or potentially lie in their physical, mental, and social aspects. There was a weak association between ward nurses’ self-assessment of oral care and problem-solving behavior. It is presumed that what the ward nurses do is not simply oral care but providing oral care as the problem-solving behavior to solve patients’ health problems by making a holistic assessment of patients. This suggests that the improvement of problem-solving behavior results in the enhancement of oral care quality. It is necessary for ward nurses to learn communication skills to confirm what patients want and to select and carry out oral care tailored for individual patients to implement high-quality oral care. Thus, future lectures and practical training sessions should not only concentrate on the manual preparation of oral care as a care, but as a nursing practice with an understanding of patients’ individual conditions. Oral care techniques, oral anatomy, and diseases should be taught so that knowledge of the oral area can be acquired to respond to individual patients. Another study reported that collaboration with dental specialists is effective for nurses to acquire knowledge and techniques of oral care and to implement oral care, thereby contributing to the change of nurses’ awareness of oral care and enhancement of oral care quality [23]. The present study showed that 83.7% of the nurses solved their problems, and 87.3% improved their awareness among the nurses who received on-site guidance and instructions from dental specialists during the past years. In this study, the nurses could feel the increased burden and receive negative feedback if they pursued high-quality oral care. Therefore, we did not ask any questions leading to the assessment of oral care quality. However, these results suggest that collaboration with dental specialists helps acquire knowledge and techniques of oral care and to change nurses’ awareness of oral care. A previous study reported that there are some benefits of nurses’ working with dental specialists as a coordinator in a team for medical care [23]. One is that nurses and dental specialists can duplicate their care by sharing patients’ goals and understanding. They can also provide oral care in consideration of patients’ general conditions and lifestyles. We therefore conclude that it is necessary to improve nurses’ abilities to select an appropriate oral care method, to implement assessment and oral care, and to efficiently solve the problems. We also conclude that an education program focusing on collaboration with dental specialists is required.

## 5. Limitations

There are several limitations to this study. First, this study focused on a single hospital in an aging community. Second, the hospital had only a small number of male nurses, which prevented us from examining differences between the genders. Third, the oral and maxillofacial surgery department in the hospital was not always available to provide oral care throughout the year since the department is open only twice a week. Fourth, because of the COVID-19 pandemic, we were unable to hold large-group discussions or study groups. In addition, access to the wards was restricted due to the cluster outbreak, which limited the scope of activities for both nurses and dental specialists. Hence, we are unable to rule out the possibility that the results of this study were influenced by regional idiosyncrasies, the features of the subject facilities, and the COVID-19 pandemic. In general, the COVID-19 pandemic has changed the conditions of fragmentation of care and comprehensiveness of medical services [24]. Therefore, care needs to be exercised in generalizing the results of this study. Considering these limitations, in the future, we would like to conduct joint research at other facilities to examine how changes in perceptions affect nursing practice and the turning points of patients.

## 6. Conclusions

In this study, dental specialists succeeded in significantly increasing nurses’ interest in and priority to oral care by intervening in the clinical practice, but failed in decreasing nurses’ burden of oral care. The present interventions were unable to decrease the nurses’ oral care burden. Suggestions from the nurses were often received regarding matters that were not included in the questionnaire. In the future, we plan to collect qualitative data by incorporating open-ended questions into the questionnaire, examine problems, and seek improvement in oral care. To provide more information for nurses, we plan to actively hold large-group lectures and practical skill training sessions, which were restricted due to the COVID-19 pandemic.

## Figures and Tables

**Table 1 healthcare-11-02425-t001:** Demographic characteristics of nurses (*n* = 136).

Category	Overall (%)
Gender	
Male	9 (6.6)
Female	127 (93.4)
Age	
20–29 years	27 (19.9)
30–39 years	22 (16.2)
40–49 years	43 (31.6)
50–59 years	36 (26.5)
60–69 years	8 (5.9)
Interest	
Strong	14 (10.3)
A little	111 (81.6)
Neutral	7 (5.1)
No interest	4 (2.9)
No idea	0(0)
Necessary	
Strong	73 (53.7)
A little	62 (45.6)
Neutral	1 (0.7)
No interest	0 (0)
No idea	0 (0)
Priority	
Very high	5 (3.7)
Relatively high	76 (55.9)
Not high	36 (26.5)
Not relatively high	19 (13.9)
No idea	0 (0)
Involvement with Oral and Maxillofacial Surgery	
Yes	49 (36.0)
No	87 (64.0)
Department	
Outpatient	20 (14.7)
Ward	109 (80.1)
Operating room	7 (5.1)

**Table 2 healthcare-11-02425-t002:** Comparison of nurse background before and after intervention.

	Intervention		
Factor	After the Interventions	Before the Interventions	*p* Value
N	136	134	
Age (%)			
20–29 years	27 (19.9)	28 (20.9)	0.662
30–39 years	22 (16.2)	22 (16.4)	
40–49 years	43 (31.6)	41 (30.6)	
50–59 years	36 (26.5)	29 (21.6)	
60–69 years	8 (5.9)	14 (10.4)	
Gender (%)			
Male	9 (6.6)	6 (4.5)	0.597
Female	127 (93.4)	128 (95.5)	
Nursing experience	19.33 (12.50)	18.75 (11.98)	0.699
Setting (%)			
Outpatient	20 (14.7)	20 (14.9)	0.999
Ward	109 (80.1)	109 (81.4)	
Operating room	7 (5.1)	5 (3.7)	

**Table 3 healthcare-11-02425-t003:** Comparison of surveys on nurses’ oral care before and after intervention.

	Intervention		
Factor	After the Interventions	Before the Interventions	*p* Value
N	136	134	
Higher Need (%)	135 (99.3)	132 (98.5)	0.621
Higher Interest (%)	125 (91.9)	99 (73.9)	<0.001 *
Higher Priority (%)	81 (59.6)	41 (30.6)	<0.001 *
Frequency, mean (SD)	2.32 (0.86)	2.32 (1.02)	1
Ideal time, mean (SD)	3.05 (0.73)	3.22 (0.96)	0.146
OHBI, Median, [IQR]	25.00 [9.00, 39.00]	25.00 [15.00, 43.00]	0.593
Oral care time, Median, [IQR]	5.00 [1.00, 30.00]	5.00 [0.00, 15.00]	0.569

* *p* < 0.05. Higher Need: strong or a little. Higher Interest: strong or a little. Higher Priority: very high or relatively high. Frequency (times): Number of oral care sessions per patient per day. Ideal time (times): Ideal number of oral care sessions per patient per day. OHBI, oral health-related caregiver burden index. Oral care time (minute): Oral care time per session. Answer (score): never (1), rarely (2), sometimes (3), quite frequent (4), nearly always (5). SD, standard deviation. IQR, Interquartile range.

## Data Availability

The data that support the findings of this study are available from the corresponding author.

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
