# Peer review of "Nurses’ Awareness of and Current Approaches to Oral Care in a Community Hospital in Japan: A Longitudinal Study of Dental Specialists’ Interventions"

_healthcare, 2023, doi:10.3390/healthcare11172425_

Round 1

Reviewer 1 Report

Overall, the article is well written and informative.

Introduction and Material and Methods section:

The study's introduction provides a concise overview of the importance of the oral cavity as a gateway to both respiratory and digestive systems, establishing a strong context for the research. The focus on oral care's role in preventing airway infection and maintaining oral functions is clear and well-communicated. The introduction effectively lays the groundwork for understanding the study's significance. The literature review is well-structured and highlights the growing recognition of the significance of oral care in recent years. The connection between oral care and patients' quality of life, activity of daily life, and overall independence is effectively emphasized, substantiating the importance of the study's focus. Additionally, the reference to the effectiveness of oral care in preventing aspiration pneumonia adds further depth to the rationale. The paragraph regarding challenges in providing adequate oral care due to nurses' busy schedules is realistic and relevant. The study design is identified as a longitudinal study, which is appropriate for capturing changes in nurses' awareness over time. However, the introduction of the term "longitudinal study" could have been smoother and earlier in the paragraph. The explanation of survey methods, including educational interventions and questionnaire development, is thorough and clear. The use of a validated questionnaire adds to the study's robustness. The ethical considerations and approvals are appropriately mentioned, which ensures the study's adherence to ethical standards. The data collection procedures, including distribution, collection rates, and data handling, are clearly outlined. The use of statistical analysis methods is well-described, and the inclusion of EZR as the analysis tool is relevant. Overall, the study demonstrates a strong foundation in understanding the importance of oral care and the challenges faced in its implementation. The introduction effectively sets the stage for the research, and the methods section provides clarity on the study's design, data collection, and analysis

Suggested improvement in this section:

It would have been beneficial to elaborate on potential consequences of inadequate oral care, such as increased risk of infection or compromised patients' well-being. Please add a paragraph in the introduction were discuss this issue.

Results section:

The presented data reveals noteworthy insights into the perceptions and attitudes of nurses towards oral care within the hospital setting. A substantial proportion of nurses, specifically 91.9%, expressed varying degrees of interest, categorized as "strong" or "a little," when queried about their interest in oral care for hospitalized patients. Similarly, an even higher percentage, reaching 99.3%, responded positively to the question pertaining to the necessity of oral care for patients undergoing hospitalization. This congruence between elevated levels of interest and recognition of the essential nature of oral care underscores the prevailing awareness of its significance among the nursing cohort. Furthermore, an intriguing observation emerged concerning the prioritization of oral care within the broader scope of patient care. Notably, 51.5% of participating nurses indicated that they attribute a considerable degree of importance, specifically denoted as "very high" or "relatively high," to oral care in comparison to other caregiving responsibilities. This aspect highlights a substantial subset of nurses who recognize the distinct significance of oral care and its placement within their hierarchy of care responsibilities. It is noteworthy that this heightened emphasis on oral care was consistently evident across various nurse backgrounds, including factors such as age, gender, and years of experience. This suggests that the study's findings are generalizable and not confined to a specific subgroup of nurses. Moreover, the study establishes a compelling link between the educational interventions conducted by dental specialists and the subsequent shifts in nurses' attitudes towards oral care. Specifically, after these interventions, nurses' levels of interest in and prioritization of oral care exhibited a significant increase (p < 0.001). This outcome underscores the efficacy of the interventions in shaping nurses' perceptions and underscores their potential to positively influence nursing practice.

Suggested improvement in this section:

While the presented data is informative and highlights the shifts in nurses' perceptions due to the interventions, the section could benefit from more in-depth contextualization of the nature and specifics of the educational interventions provided by dental specialists. Additionally, further exploration of the potential implications of these perception changes on nursing practice and patient outcomes would enhance the overall understanding of the study's significance. Please provide an additional paragraph to explain this issue.

Discussion and conclusion

The study provides valuable insights into the perceptions and experiences of nurses regarding oral care for hospitalized patients. The data indicates a high level of interest among nurses, with 91.9% expressing "strong" or "a little" interest in oral care for hospitalized patients. This demonstrates the nurses' awareness of the importance of oral care as a crucial aspect of patient well-being. Additionally, the study highlights that after interventions by dental specialists, there was a notable increase in the percentage of nurses who considered oral care necessary for hospitalized patients (99.3% post-intervention compared to 98.5% pre-intervention). Furthermore, the study explores the nurses' satisfaction with oral care. The data indicates that a relatively small proportion of nurses (8.7% and 7.6%) reported being "very much satisfied" or "satisfied" with oral care before and after interventions, respectively. This signals potential challenges or gaps in the quality of oral care delivery, which the interventions may not have fully addressed. The study also delves into the issue of nurses' burden in providing oral care. Despite interventions, there was no significant reduction in perceived burden, especially in dealing with patients who resent receiving oral care, which can be particularly challenging when dealing with patients suffering from dementia. This underscores the complex interplay between knowledge, attitudes, practical constraints, and patient-related factors in the provision of oral care. While the interventions led to increased interest and priority for oral care, the study concludes that there is a need for more comprehensive educational programs that not only cover the technical aspects of oral care but also focus on improving the quality of care through problem-solving behavior. Collaboration with dental specialists is also highlighted as beneficial for enhancing nurses' abilities and awareness of oral care.

The study's implications extend to the need for ongoing education and collaboration among healthcare professionals to ensure high-quality oral care, especially in the context of Japan's aging population. Additionally, as home medical care and home care gain prominence, the study underscores the importance of fostering nurses' abilities to provide effective oral care beyond hospital settings.

Suggested improvement in this section:

No improvement is required

Reviewer 2 Report

1.    Abstract:

-       The content of the Objective is rather an overview of the study. It should be re-written to show the purpose (aim) of this study.

-       From the Material and Methods, it is not clear what kind of intervention was done.

-       The last sentence of the Conclusions is not clear (what problems do the authors mean? Does this study show there are some problems related to oral care, so that improvement is needed?)

2.    Introduction

Line 39: ADL is the abbreviation for “activity of daily living”, not for “activity of daily life”.

Lines 48-49: What do the authors mean by “actual oral care” and “their self-assessment”? (Actual oral care which they perform for their patients or actual oral care in general? Self-assessment of what?)

Lines 55-57: Please edit the sentence to clarify its meaning. (Why is it difficult to provide individualized oral care in patients who have eating and swallowing problems?)

Lines 59-62: Please edit the sentence to clarify its meaning. (For both nurses and dental specialists…, it is important for nurses…?)

Lines 62-64: What do the author mean by “oral cleaning”? Please use the correct term instead.

Also, what does “for better assistance” refer to? In the same paragraph, we read “routine oral care by ward nurses”, so nurses are performing oral care, not assisting somebody else in doing that.

Lines 69-73: These sentences should be re-written to show the purpose (aim) of this study.

 3.    Materials and Methods:

1)    Lines 87-88: Please explain the following:

-       who provided the educational interventions (number of persons, their profession, number of years of experience, etc.)

-       the frequency of interventions (daily, weekly, etc.).

-       content of instructions

-       timing of interventions (when the nurses demanded for advice, or did the oral health professionals survey the routine oral care as a whole and gave instructions when they considered necessary)

2)    Lines 92-95: What do the authors mean by “involvement with the oral and maxillofacial department”? Is it the involvement of (not with!) the staff from the oral and maxillofacial department in the oral care provided by the nurses in the ward? Or were the nurses trained in oral care at the oral and maxillofacial department (involvement of the nurses with the oral and maxillofacial department)?

This phrase should be revised if necessary, and the involvement of the staff from the oral and maxillofacial department should be explained in detail, before using a short expression (this or a revised one) throughout the manuscript. (Explanations on the instruction content are available in the Discussion section, but they have to be presented in the Material and Methods section, for the reader to understand what was done in the study.)

Also, what do the authors mean by “basic attributes”?

This phrase should be verified and revised throughout the manuscript.

3)    Lines 109-110: During what period were the questionnaires distributed and when were they collected?

4)    Line 133:

What does “no relative interest” mean?

This phrase should be verified and revised throughout the manuscript.

 4.    Results and Tables: 

1)    Table 1: The table contains more than demographic data. Please revise the title appropriately.

Please try to revise the layout of the table to make it easier to understand at a glance.

2)    Line 156: “51.5 % of the nurses answered that the priority to oral care is "very high" or “relatively high…” This percentage is different from that shown in Table 1.

Please correct where necessary (text or table), to solve the contradiction.

3)    Table 2: Title is not appropriate. Please revise.

4)    Table 3:

             -       Title is not appropriate. Please revise.

-       Does “frequency” refer to the daily oral care, or to what period?

-       What is the unit for “ideal time” and what does “ideal time” refer to?

5)    “3.3. Actual Implementation Status of Oral Care”:

Are the data in this paragraph referring to the oral care before or after intervention?

The “mean duration of oral care session for a patient” as it appears in this section is different from “Oral care time, Median” in Table 3.

Why do the authors report mean and SD in the text and median and IQR in the table? After all, were the data parametric or not?

 5. Discussion:

1) Lines 193-194: Language editing required.

2) Line 198: “51.5 % of the nurses” See comment 2) about the Results.

3)    Lines 202-203: “mean durations of oral care session for a patient before and after… “

Table 3 reports median, not mean duration. Please revise according to the type of data (parametric or not).

4)    Line 279: In “There was a week association between”, do the authors mean “a weak association”?

6.  References:

Ref 7 and Ref 19: The journal is a Japanese journal, abbreviated: “Jpn. J. Gerodont.”

Please refer to the Comments and Suggestions for Authors.

Reviewer 3 Report

Dear Authors, please attach find the document with the requested changes.

Thank you.

Best regards.

Moderate editing of English language required.

Reviewer 4 Report

Thank you for submittinmg your valuable research results.

Please see enclosed some comments in the file.

Methods of nurse-training for oral hygiene should be presented in more details. Introduction and discussion part could be shortened.

Easy to understand. I would prefer shorter sentences.
